# CoO Nanozymes with Multiple Catalytic Activities Regulate Atopic Dermatitis

**DOI:** 10.3390/nano12040638

**Published:** 2022-02-14

**Authors:** Mao Mao, Xuejiao Guan, Feng Wu, Lan Ma

**Affiliations:** 1School of Life Sciences, Tsinghua University, Beijing 100084, China; larva1949@126.com (M.M.); gxj920309@126.com (X.G.); wu8721@126.com (F.W.); 2Institute of Biopharmaceutical and Health Engineering, Tsinghua Shenzhen International Graduate School, Tsinghua University, Shenzhen 518055, China; 3State Key Laboratory of Chemical Oncogenomics, Tsinghua Shenzhen International Graduate School, Tsinghua University, Shenzhen 518055, China; 4Institute of Biomedical Health Technology and Engineering, Shenzhen Bay Laboratory, Shenzhen 518055, China

**Keywords:** nanozyme, CoO, enzyme-like catalytic activity, atopic dermatitis

## Abstract

Herein, we prepared CoO nanozymes with three types of enzyme catalytic activities for the first time, which have SOD-like, CAT-like, and POD-like catalytic activities. This is the first study to report the preparation of CoO nanoparticles with three types of enzyme catalytic activities by the one-pot method. By modifying the surface of CoO nanozymes with a carboxyl group, its biocompatibility enhanced, so it can be used in the field of life sciences. In vitro cytotoxicity and anti-H_2_O_2_-induced ROS experiments proved that CoO nanozymes can protect HaCaT cells against ROS and cytotoxicity induced by H_2_O_2_. In addition, an atopic dermatitis (AD) mouse model was established by topical application of MC903, which verified the anti-inflammatory effect of CoO nanozymes on the AD mouse model. Traditional drugs for the treatment of AD, such as dexamethasone, have significant side-effects. The side-effects include skin burns, telangiectasias, and even serious drug dependence. CoO nano-enzymes have a low cytotoxicity and its multiple enzyme-like catalytic activities can effectively protect cells and tissues in ROS environments, which proves that CoO nano-enzymes have high application potential in the field of anti-inflammation.

## 1. Introduction

Nanozymes are a new type of functional nanomaterial that have enzyme-like catalytic activity. Compared with natural enzymes, nanozymes have many advantages, including higher stability, tolerance to harsh environments, and relatively low production costs [1,2,3,4,5,6]. Natural enzymes have poor stability, low yield, and significant production costs, while their stringent requirements on the environment largely limit the use of natural enzymes in various fields. Nanozymes are produced in a simpler and more economical way than natural enzymes [4]. More importantly, the catalytic activity of nanozymes can be enhanced by adjusting their size, morphology, and composition, even comparable to natural enzymes [7,8,9,10]. In addition, nanozymes can have some properties that natural enzymes cannot, such as large specific surface area, catalytic activity that can be regulated, and response to external stimuli [4]. Therefore, nanozymes have received great attention in recent years, and they have also been greatly used in biosensor development, medical diagnosis, treatment, and tissue engineering.

According to the properties of nanozymes, they can be roughly divided into two categories: the first is oxidoreductase, including peroxidase (POD) [11], haloperoxidase, catalase (CAT), glucose oxidase enzymes, sulfite oxidase, superoxide dismutase (SOD), laccase, monooxygenase, CO oxidase, and ferritin ferrous oxidase [12]; the second is different hydrolases (phosphatase, phosphotriesterase, and dehydratase), as well as proteases, endonucleases, DNases, NO synthases, etc. [6,13,14]. CAT is a conjugated enzyme with iron porphyrin as a prosthetic group [15], which can catalyze hydrogen peroxide (H_2_O_2_) to oxygen and water, thereby protecting tissues from excess H_2_O_2_ damage [16]. So far, a series of metal-related nanozymes, such as platinum (Pt) [17], gold (Au) [18], CeO_2_ [19], and Mn_3_O_4_ [20], have been demonstrated to show CAT-like activity. Peroxidases catalyze the oxidation of organic substrates with H_2_O_2_ as an electron acceptor, thus decomposing H_2_O_2_ and effectively eliminating the toxicity of phenolic (Fe_3_O_4_) nanoparticles with specific properties similar to HRP [21]. Now, a series of nanomaterials have been used as POD mimics, including metal materials [22], metal oxides [23], conducting polymers [24], metal–organic frameworks [25], carbon nanomaterials [26], and single-atom catalysts [27]. Oxidative stress, including increased concentrations of reactive oxygen species (ROS), is considered to be an important factor in aging and disease [28]. ROS refers to the reduction products of oxygen in the body, including oxygen radicals (such as O^2•−^, ^•^OH, and HO^2•^) and certain nonradical oxidants (such as ozone, H_2_O_2_, and hypochlorous acid) [29]. SOD has been selected as a useful tool for anti-oxidation and anti-aging because it can convert superoxide anion radicals to H_2_O_2_ and O_2_ [30]. Many nanomaterials have been shown to be SOD mimetics, such as Mn_3_O_4_ [31], Au [32], MnO_2_ [33], and CeO_2_ [34].

The enzyme-like catalytic activity of nanozymes depends on the size, shape, and structure of nanoparticles, and the shape of nanozymes is affected by coatings, charges, and external electric fields [35]. Different methods for preparing nanozymes can obtain nanozymes with different properties. For example, low-cost and size-controllable nanocrystals can be obtained by hydrothermal and solvothermal methods [36,37]. A series of spinel-type nanocrystals were prepared by the solvothermal method using ethylene glycol as a solvent, and these nanozymes were used for the detection of H_2_O_2_ [38,39]. Chemical reduction can control the morphology and particle size of the obtained nanoparticles by changing the amount of reactants, the type of reducing agent, and the reaction temperature [40]. This approach has been used to prepare gold nanoparticles with peroxidase-like catalytic activity [41]. In the sol–gel method, the crystallinity, morphology, magnetic properties, and other properties of the prepared nanozymes are controlled by selecting a suitable complexing agent, changing the concentration and type of chemical additives, and reaction temperature [42]. Platinum nanoparticles polyaniline (PAni) hydrogel heterostructures were prepared by this method [43]. In addition to the above synthetic methods, nanozymes can also be prepared by co-precipitation, electrochemical deposition, polymerization, polycondensation, and other methods [35].

Among nanozymes, metal oxide nanozymes have a high surface energy and high specific surface area, and have been considered as artificial enzymes with great application potential for decades. The most commonly reported metal oxide nanozymes, such as CeO_2_, Fe_2_O_3_, Fe_3_O_4_, Co_3_O_4_, Mn_2_O_3_, and Mn_3_O_4_, have a variety of enzyme catalytic activities [44]. Furthermore, they also exhibit many unique properties, such as magnetism, fluorescence quenching, and dielectric properties [45]. However, unmodified metal oxide nanozymes have some disadvantages in biomedical applications. For example, they may show extremely poor stability under physiological conditions, and even accelerate the production of harmful free radicals [46]. In addition, improper surface ligand coating may lead to failure of drug release control [47].

Cobalt oxide, especially Co_3_O_4_ nanozymes, have CAT-like, POD-like, and SOD-like catalytic activities [48]. Co_3_O_4_ nanozymes are widely used in cancer therapy, antiviral, magnetic resonance imaging (MRI), and targeted drug delivery [49,50,51]. At present, many studies have reported that CoO nanozymes with POD-like activity prepared by a variety of synthetic methods have extremely high application value in biosensing [52,53,54,55]; CoO nanostructures have the advantages of high stability, large specific surface area, and easy redox [56,57]. However, pure CoO nanoparticles show poor electrical conductivity and tend to self-aggregate to cover active sites. Therefore, many studies have sought to further improve the catalytic performance of CoO nanoparticles. For example, combining with carbon materials can effectively improve the electron transfer efficiency and the dispersion of CoO nanoparticles [58,59]. The introduction of CoO nanodots into nitrogen-doped porous carbon increases the electron density on the carbon surface, thus enhancing both radical and nonradical catalytic processes [60]. In addition, functional doping with metals or metal oxides is also an effective method. For example, the homogeneous doping of SiO_2_ onto CoO nanorods significantly enhanced the latter’s intrinsic peroxidase-like activity [61]. Currently, there are only a few works on CoO-based catalysts as nanozymes. Therefore, it is necessary to study novel CoO nanozymes with special properties.

Currently, topical glucocorticoids, including dexamethasone, are the first-line treatment for AD. Its advantages are fast digestion and absorption, and can effectively control the disease, but the side-effects are also obvious. Dexamethasone may cause local skin atrophy, telangiectasia, and skin burns, and can lead to severe drug dependence.

There are many studies linking oxidative stress with atopic dermatitis, and studies have now incorporated antioxidants into the treatment strategy of AD [62]. Nanozymes can regulate the ROS level by mimicking the activity of natural antioxidant enzymes, and have been widely used in areas such as cell protection and anti-inflammatory disease [63]. In this study, we prepared CoO nanozymes with SOD-like, CAT-like, and POD-like catalytic activities by the one-pot method. After surface modification of CoO nanozymes, its biocompatibility has been greatly improved. Then, it was applied to the regulation of ROS in atopic dermatitis.

## 2. Materials and Methods

### 2.1. Materials

ROS, CAT, SOD (WST-8 method), the CCK-8 detection kit, and antioxidant NAC were purchased from Biyuntian Company (Shanghai, China). The POD detection kit, MC903, bovine serum albumin (BSA), cobalt acetylacetonate, dodecanol, octadecene, oleic acid, and oleylamine were purchased from Sigma (Shanghai, China). H_2_O_2_, ethanol, and isopropanol were purchased from Shenggong Company (Shenzhen, China). Rabbit anti-IgG and TSLP antibody (ab188766) were purchased from Abcam (Cambridge, UK). EDTA antigen retrieval solution, an immunohistochemistry kit, TB staining solution, and glacial acetic acid were purchased from Servicebio (Wuhan, China). Xylene and neutral gum were purchased from Sinopharm (Beijing, China). Dexamethasone (compound dexamethasone acetate gel) was purchased from Jinri Pharmaceutical Company (Xiamen, China).

### 2.2. Characterizations

The surface topography and crystallographic structure of CoO nanoparticles were acquired by transmission electron microscopy (TEM, F30, FEI, Hillsboro, OR, USA) and field-emission scanning electron microscopy (SEM, SU8010, HITACHI, Tokyo, Japan). The hydrodynamic size and surface potential of the product were measured by dynamic light scattering (DLS) using a Zetasizer (Nano ZS90, Malvern Instruments, Shanghai, China). UV–vis–NIR absorbance spectra of CoO NPs were measured using a spectrophotometer (UV-1800, Shimadzu, Kyoto, Japan). The crystalline form of the nanostructure was characterized by X-ray diffraction (XRD) using an X-ray diffractometer (XRD-7000, Shimadzu, Kyoto, Japan) with CuKα radiation (λ = 1.5406 Å).

### 2.3. Preparation

Synthesis of CoO nanoparticles: CoO nanoparticles were prepared by the one-pot method. Generally, 0.716 g of cobalt acetylacetonate, 20 mL of ODE, 1.342 mL of dodecanol, 0.6323 mL of OA, and 1.9741 mL of OAM were mixed uniformly. It was exhausted under N_2_ for 30 min. The reaction was carried out at 270 °C for 1 h. After the reaction was completed, it was cooled to room temperature, purified, and dispersed in chloroform for later use.

Carboxyl-modified CoO nanoparticles: CoO nanoparticles were modified using PMA-ODE. Typically, 0.05 g of CoO nanoparticles and 0.25 g of PMA-ODE were dispersed in chloroform. After drying, 5 mL of H_2_O (containing 1 mL ammonia) was added, and CoO was dispersed into water by ultrasound to obtain hydrophilic CoO nanozymes, which was centrifuged and dispersed into water for later use.

Determination of SOD-like, CAT-like, and POD-like catalytic activities of CoO nanozyme: The enzyme-like catalytic activities of CoO nanozymes were detected using SOD, CAT, and POD kits, respectively, as shown in the kit instructions.

### 2.4. In Vitro and In Vivo Experiment

Animals and cells: Balc/c mice were obtained from the Laboratory Animal Center, Guangdong Province, China. All of the animals were maintained in cages in a temperature-controlled environment, with a 12 h light-dark cycle and free access to food and fresh water. HaCaT cells were cultured in DMEM containing 10% FBS and antibiotics (100 units/mL penicillin and 100 μg/mL streptomycin) and incubated at 5% CO_2_ and 37 °C in humidified air.

Cytotoxicity of CoO nanozymes on HaCaT Cells: The cytotoxicity of CoO nanozymes was measured by the CCK-8 assay. Briefly, cells with a cell density of 5 × 10^3^/well were seeded in a 96-well plate and incubated overnight. Then, cells were pretreated with CoO nanozymes at different concentrations (2.5, 5, 10, 20, 40, and 80 μg/mL) for 24 h. CCK-8 solution was added into the medium and incubated for 4 h. The absorbance was measured at 450 nm (Spectramax M2e, Molecular devices, Sunnyvale, CA, USA).

Cell viability and ROS in HaCaT cells induced by H_2_O_2_: Cell viability was measured by the CCK-8 assay. In short, cells with a cell density of 5 × 10^3^/well were seeded in a 96-well plate and incubated overnight. Next, the cells were exposed to different concentrations (500, 600, 700, 800, and 900 μM) of H_2_O_2_ for 24 h. CCK-8 solution was added to the medium and incubated for 4 h. The absorbance was measured at 450 nm. Intracellular ROS were measured by the DCF-DA fluorescence assay. Cells were grown in a 96-well plate with a transparent bottom with black wells for 24 h. Then, cells were treated with 700 μM of H_2_O_2_ for 1 h. Subsequently, the cells were washed twice with phosphate-buffered saline (PBS) and stained with 20 μM of DCF-DA for 30 min. The fluorescence was read at 485 (excitation)/535 (emission) nm (Spectramax M2e, Molecular devices).

Protective effect of CoO nanozymes on H_2_O_2_-induced HaCaT cells: Cell viability was measured by the CCK-8 assay. Generally, cells with a cell density of 5 × 10^3^/well were inoculated in a 96-well plate and incubated overnight. Next, the cells were pretreated with different concentrations (2.5, 5, 7.5, 10, and 20 μg/mL) of CoO nanozymes and NAC, and the final concentration was 15 μg/mL. After 24 h, cells were exposed to 700 μM of H_2_O_2_ for 24 h. After the reaction, CCK-8 solution was added to the medium and incubated for 4 h. The absorbance was measured at 450 nm. Flow cytometry analysis was used to monitor the ROS level in HaCaT cells by the DCFH-DA detector. HaCaT cells were inoculated into 6-well plates at a density of 10^6^ cells per well and cultured for 24 h. Then, the cells were pretreated with CoO nanozymes and NAC at different concentrations (1.25, 2.5, 5, 10, and 20 μg/mL) to a final concentration of 15 μg/mL. After 6 h, the cells were exposed to 700 μM of H_2_O_2_ for 1 h. Next, the cells were washed twice with phosphate-buffered saline (PBS) and stained with 20 μM of DCF-DA for 30 min. Finally, the fluorescence intensity was detected by flow cytometry.

Preparation of nanozyme hybrid gel. The gel was prepared by mixing 3.5 g of methylcellulose with 10 mL of glycerol and adding 90 mL of deionized water, which was dispersed at room temperature for 24 h and stored at 4 °C. The aqueous solution of CoO nanozymes and the prepared gel were mixed in a ratio of 1:1 to form a 20 and 40 μg/mL CoO nanozyme gel, and stored at 4 °C for backup.

Design of animal experiment for AD and collection of tissue samples: To establish an AD model in mice, MC903 was repeatedly applied into the ears of Balb/c mice [64,65,66]. The Balb/c female mice were randomly divided into the control group, the model group, the dexamethasone group, and the 20 and 40 µg/mL CoO nanozyme groups, with 4–5 mice in each group. An amount of 4 nM of MC903 (dissolved in 20 μL of absolute ethanol) was smeared on both sides of the ears for 5 days in the model group, CoO nanozyme group, and dexamethasone group daily, while 20 μL of anhydrous ethanol was smeared on the mice for 5 days in the control group. Following a period of 10 days, the frequency of MC903 treatment was changed to once every two days. Meanwhile, 4 h after MC903 treatment, 20 μg/mL of CoO nanozyme gel and 40 μg/mL of CoO nanozyme gel and compound dexamethasone gel were smeared on the CoO nanozyme group and dexamethasone group, respectively. The swelling degree of the ears of the mice was observed every day, and the thickness of the ears of the mice was measured and recorded with a vernier caliper before the mice were euthanized.

The visceral organs of mice in different treatment groups were collected and immersed in 4% paraformaldehyde fixed solution for follow-up pathological section study. The ear tissues of mice were cut off by surgical scissors and some of them were soaked in 4% paraformaldehyde fixed solution for follow-up pathological section study.

## 3. Results and Discussion

### 3.1. Properties of CoO Nanoparticles

Figure 1A shows the TEM image of hydrophobic CoO nanoparticles. From the image, we can see that the “cluster”-like nanoparticles were prepared, and their particle size and morphology distribution were uniform. The particle size of CoO nanoparticles was analyzed by Nanomeasure software, and the particle size of hydrophobic CoO NPs was between 25 and 30 nm. Figure 1B shows the XRD spectrum of CoO nanoparticles. Compared with the standard card, it proves that the nanoparticles we prepared were CoO. In addition, Appendix A describes the results of scanning electron microscopy (SEM) and energy dispersive X-ray spectroscopy (EDS) analysis. As can be seen from Appendix A, the obtained CoO nanoparticles were uniformly distributed with a rough surface. EDS analysis is often used to determine the elements present in reaction products. The EDS spectrum consisted of distinct peaks corresponding to Co, O and C, respectively (Appendix A). The large and intense Co and O peaks originated from the CoO nanoparticles. The carbon peaks came from the conductive glue used to make the samples for EDS analysis.

CoO nanoparticles form a special “cluster” morphology because of the insufficient temperature during the CoO nanoparticles’ growth; lower temperatures will cause the crystals to crack and form “cluster” crystals. As the reaction temperature increases, the crystallinity of the obtained CoO nanoparticles increases. When the reaction temperature is increased to 320 °C, CoO nanoparticles with complete crystals can be obtained (Appendix A), but their enzyme-like catalytic activity will be affected. Appendix A shows images of different concentrations of CoO nanoparticles reacting with TMB for 10 s. The concentration of CoO nanoparticles was 80 mg/mL, and 0, 10, 20, 50, and 100 μL of CoO nanoparticle solutions were added to 1 mL of TMB solution, respectively. As shown in Appendix A, CoO nanoparticles prepared at 270 °C reacted with TMB for 10 s. The TMB solution turned blue, and the color of the TMB solution gradually deepened as the concentration of CoO nanoparticles increased. In Appendix A, after the CoO nanoparticles prepared at 320 °C were reacted with TMB for 10 s, the TMB solution turned blue-green, and the color of the solution also deepened with the increase in the concentration of CoO nanoparticles. The solution was blue-green because the discoloration reaction had not been completed, and the solution would eventually turn blue when the reaction time is increased. It can be seen from the figure that the CoO nanoparticles prepared at 270 °C had a higher POD-like catalytic activity. More importantly, we did not observe an SOD-like catalytic activity on CoO nanoparticles prepared at 320 °C. Some of the literature mentions that flower-like nanozymes will show a higher enzyme-like catalytic activity due to their larger specific surface area [53]. Based on this, we speculate that the change in the catalytic activity of the enzyme-like enzymes after the change in the morphology of CoO nanozymes is also derived from the change in the specific surface area and the opening of surfactant sites.

### 3.2. Properties of Carboxyl-Modified Hydrophilic CoO Nanoparticles

Hydrophobic CoO nanoparticles were functionalized by PMA-ODE, then hydrophilic CoO nanoparticles with carboxyl groups were obtained (Figure 2A), and their hydrated particles were about 91 nm (Figure 2B). It is obvious from Figure 2C that CoO nanoparticles were dispersed in chloroform and water, respectively, before and after modification.

Hydrophobic CoO nanoparticles cannot be directly used in the field of life sciences, due to their poor biocompatibility. Thus, we used amphiphilic polymers to modify CoO nanoparticles, which greatly enhances the biocompatibility of CoO nanoparticles and enables them to be uniformly dispersed in water. This method is very simple and can be used to modify a variety of hydrophobic materials, with a certain degree of universality.

### 3.3. Enzyme-like Activity Determination of CoO Nanozymes

The SOD, CAT, and POD kits were used to test the enzyme-like catalytic activity of CoO nanozymes, and it was found that CoO had three types of enzyme catalytic activities. The inhibition rates of CoO nanozymes with concentrations of 5, 10, 20, and 30 μg/mL to O_2_^•−^ were 5.56, 28.15, 54.46, and 68.43%, respectively. The CAT-like and POD-like catalytic activities of CoO nanozymes were 19.01 and 18.47 U/mL, respectively. The specific operation process and results can be found in the Appendix A.

CoO nanozymes with three types of enzyme catalytic activities have not been reported yet, and most CoO nanozymes only have oxidase-/peroxidase-like catalytic activity [53]. A single enzyme-like catalytic activity greatly limits the application of CoO nanozymes in the biomedical field, so CoO nanozymes with multiple types of enzyme-like catalytic activities broaden their application scenarios. On the other hand, a single class of enzymatic catalytic activity, such as the decomposition of H_2_O_2_ by POD, cannot eliminate the oxidative stress caused by ROS very well. ROS include oxygen radicals (such as O^2•−^, ^•^OH, and HO^2•^) and some nonradical oxidants (such as ozone, H_2_O_2_, and hypochlorous acid), which are not completely eliminated by a redox catalytic reaction. CoO nanozymes with the catalytic activities of CAT, POD, and SOD enzymes are undoubtedly more advantageous in eliminating ROS. We speculate that the obtained CoO nano-enzyme has a variety of enzyme-like catalytic activities because of its special morphology, and the specific catalytic mechanism needs further study.

### 3.4. Cytotoxicity of CoO Nanozymes

We first examined the effect of a range of concentrations of CoO nanozymes on the cell viability of HaCaT cells (Figure 3A). HaCaT cells were incubated with a gradient concentration (2.5–80 μg/mL) of CoO nanozymes for 24 h, and the corresponding absorbance was detected by CCK-8. The cell viability of the control group was defined as 100%. When 2.5, 5, 10, 20, 40, and 80 μg/mL of CoO nanozymes were added, the HaCaT cell viability was 98.93, 94.70, 97.45, 96.13, 96.23, and 90.44%, respectively; the survival rate of cells treated with a low dose of CoO nanozymes was more than 90%, indicating that a low dose of CoO nanozymes had no significant effect on the viability of HaCaT cells. The cytotoxicity of CoO nanozymes was low, which makes further research possible.

### 3.5. Inhibitory Effect of CoO Nanozymes on Cytotoxicity Induced by H_2_O_2_

H_2_O_2_ is a common ROS inducer of HaCaT cells and has a cytotoxic effect. High concentrations of H_2_O_2_ can significantly reduce the cell viability of HaCaT cells [67], and we used H_2_O_2_ at a concentration of 700 μM, which can significantly increase the ROS level of HaCaT cells. The antioxidant NAC was used in the control group. The cytotoxicity of HaCaT cells was induced by 700 μM of H_2_O_2_; after pretreatment with NAC, the cytotoxicity of HaCaT cells was 88.20%. The effect of CoO nanozymes on the viability of HaCaT cells induced by H_2_O_2_ is shown in Figure 3B. The cells were pretreated with 2.5, 5, 7.5, 10, and 20 μg/mL of CoO nanozymes. After adding H_2_O_2_, the cell viability was 85.69, 98.15, 102.30, 100.65, and 95.58%, respectively. The survival rate was higher than the result of NAC pretreatment, and within a certain range, the higher the concentration of CoO nanozymes, the stronger their inhibitory effect on H_2_O_2_-induced cytotoxicity.

### 3.6. Inhibitory Effect of CoO Nanozymes on ROS Production Induced by H_2_O_2_

The DCFH-DA probe was used to detect the effect of gradient concentrations of CoO nanozymes on the ROS induced by H_2_O_2_ in HaCaT cells (Figure 3C). The results of flow cytometry showed that CoO nanozymes had no significant effect on the ROS level of HaCaT cells without H_2_O_2_ induction. Compared with the H_2_O_2_ treatment group, the fluorescence intensity of HaCaT cells pretreated with NAC or a gradient concentration of CoO nanozymes decreased significantly, indicating that ROS were partially eliminated by NAC or CoO nanozymes. With the increase in CoO nano-enzyme concentration, the more the fluorescence intensity decreased, the more ROS were cleared. Among them, the scavenging effect of 10 μg/mL of CoO nanozymes on H_2_O_2_-induced ROS was strongest, which was better than that of 15 μg/mL of antioxidant NAC. The above results indicate that CoO nanozymes can reduce the ROS induced by H_2_O_2_ in a dose-dependent manner.

### 3.7. In Vivo Cytotoxicity of CoO Nanozymes

In order to verify the safety of CoO nanozymes in mice, after modeling, mice in different treatment groups were dissected, and the heart, liver, spleen, lung, and kidney were taken out for HE staining (Figure 4). Compared with the control group, the histological state of the main organs of the mice in the CoO nanozyme treatment group was good, and there was no obvious difference between two groups. All above experimental results showed that CoO nanozymes have a good ability to regulate H_2_O_2_-induced intracellular ROS and cytotoxicity, and can protect HaCaT cells against H_2_O_2_-induced ROS and cytotoxicity. Furthermore, the CoO nanozymes at the experimental concentration have no obvious damage to the important organs of mice, which proves that CoO nanozymes have low toxicity.

### 3.8. Relieve the Pathological Symptoms of the Ear Skin of AD Mice

We observed that in the MC903 group (Figure 5A), the ears of the mice developed erythema and edema gradually, and the average thickness increased significantly with dry and crusted phenotypes. A schematic of the experimental design for the ear skin damage model and subsequent CoO NPs administration is shown in Figure 5D. After CoO nanozyme application for 10 days, CoO nanozymes obviously attenuated MC903-induced AD severity and the ear thickness (Figure 5E). Repetitive MC903 treatments induced epidermal thickness and hyperplasia, which was reversed by the application of CoO nanozymes (Figure 5B,F). Furthermore, toluidine blue staining identified that MC903-increased mast cell numbers can be inhibited by CoO nanozyme treatment (Figure 5C,G).

Dexamethasone is often used in the treatment of skin inflammation, but there are also serious side-effects such as drug dependence and skin burns. Comparing CoO nanozymes with dexamethasone in the study, after treatment with dexamethasone and CoO nanozymes, the ear inflammation of AD mice significantly reduced and the inflammation was relieved effectively. The therapeutic effect of the simple application of CoO nanozymes is not inferior to that of mature drugs, which fully proves the application potential of CoO nanozymes in the field of anti-inflammation.

CoO nanoparticles have high stability, large specific surface area, and are easy to redox [57,58]. However, in previous studies, CoO nanoparticles have often been doped with other substances as part of the catalyst. Few studies have used CoO nanoparticles alone as catalysts to participate in the reaction, and most of the related enzyme-like catalytic activities were peroxidase-like or oxidase-like [53,54,55,56]. In this paper, we prepared CoO nanoparticles with a “cluster” morphology, and the morphology of CoO nanoparticles can be changed by changing the reaction temperature. The higher the temperature, the higher the crystal integrity of the obtained CoO nanoparticles with better crystallinity. We observed that the CoO nanozymes possessed three types of enzyme-like catalytic activities at the same time, which is suspected due to its unique “cluster”-like structure. The “cluster” structure exposes active sites, but the specific catalytic mechanism is still lacking, while further research is needed. On the other hand, according to the enzyme-like catalytic properties of CoO nanozymes, we applied it to the treatment of atopic dermatitis mediated by oxidative stress, and achieved a similar efficacy to dexamethasone. However, if the catalytic activities of SOD and POD of CoO nanozymes can be continuously enhanced, its anti-inflammatory effect will be better and the application scenarios will be more abundant.

## 4. Conclusions

ROS are one of the important inducements of atopic dermatitis, and the elimination of ROS can effectively relieve the symptoms of atopic dermatitis. Nanozymes are widely used in the field of anti-inflammatories due to their excellent enzyme-like catalytic properties. In this study, we prepared CoO nanoparticles with a “cluster” morphology and uniform particle size, modified by carboxyl groups to improve their biocompatibility, and applied them to the treatment of atopic dermatitis in mice. The experimental results proved that the prepared CoO nanozymes have SOD-like, CAT-like, and POD-like enzyme catalytic activities; this is the first report so far. Further cell experiments demonstrated that CoO nanozymes can significantly inhibit the ROS and cytotoxicity induced by H_2_O_2_ and improve the survival rate of HaCaT cells. Animal experiments have proven that CoO nanozymes have no obvious side-effects on healthy tissues, which can safely and effectively relieve the swelling of the ears of AD mice, inhibit the infiltration of mast cells in the ear skin, and show a certain degree of relief of pathological symptoms.

## Figures and Tables

**Figure 1 nanomaterials-12-00638-f001:**
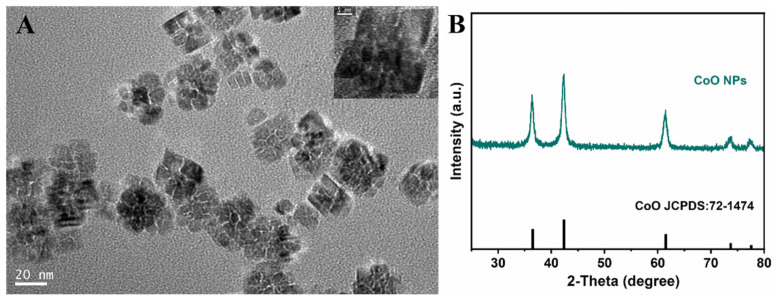
(**A**) The TEM image of hydrophobic CoO nanoparticles prepared at 270 °C. (**B**) The XRD spectrum of CoO nanoparticles.

**Figure 2 nanomaterials-12-00638-f002:**
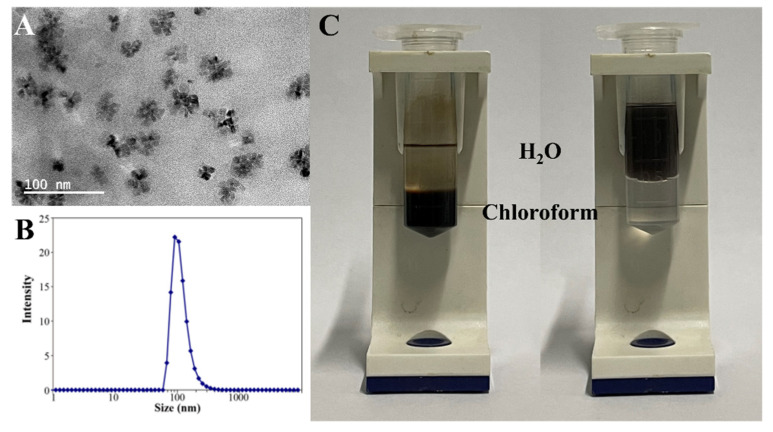
(**A**) The TEM image of hydrophilic CoO nanoparticles. (**B**) DLS spectrum of hydrophilic CoO nanoparticles. (**C**) The image of CoO nanoparticles solubility before and after modification.

**Figure 3 nanomaterials-12-00638-f003:**
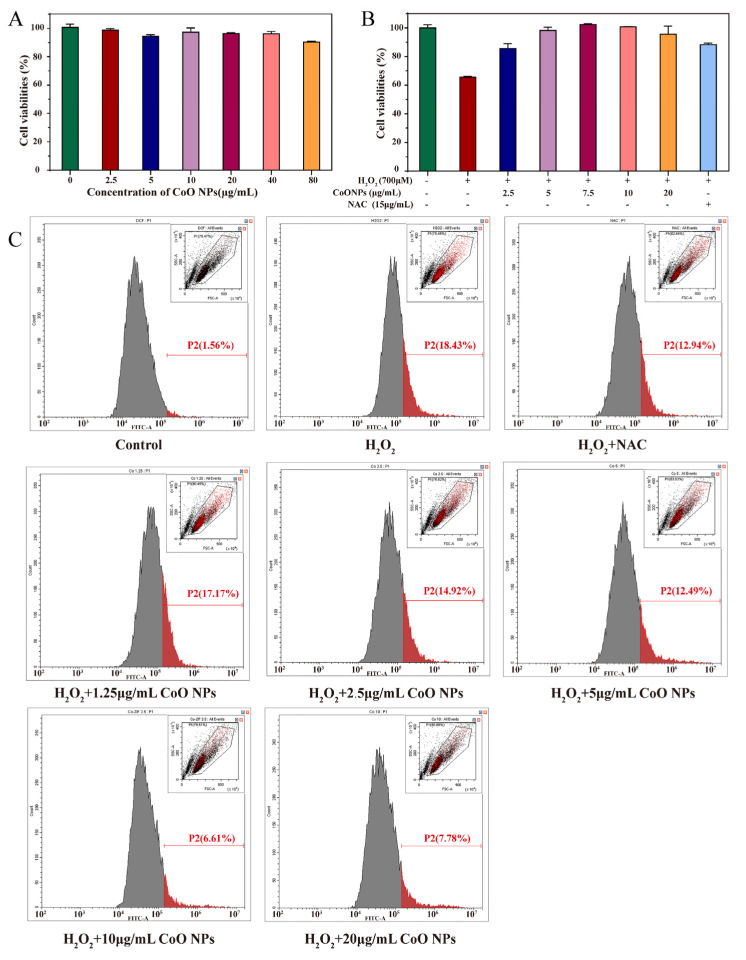
(**A**) The effect of CoO nanozymes on HaCaT cell viability. (**B**) The effect of CoO nanozymes on the cell viability of HaCaT cells induced by H_2_O_2_. (**C**) Inhibition of CoO nanozymes on ROS generation induced by H_2_O_2_ using flow cytometric analysis. The DCFH-DA probe was used to detect the effects of different gradient concentrations of CoO nanozyme on ROS induced by H_2_O_2_ in HaCaT cells. The histogram and dot plot (in the upper right corner) results represents that the CoO nanozyme had no significant effect on ROS levels in HaCaT cells in the absence of H_2_O_2_. When H_2_O_2_ was added, the level of ROS in HaCaT cells pretreated with NAC/CoO nanozyme was significantly reduced. As the concentration of CoO nanozyme increased, more ROS were eliminated. Among them, 10 μg/mL CoO nanozyme had the strongest scavenging effect on H_2_O_2_-induced ROS.

**Figure 4 nanomaterials-12-00638-f004:**
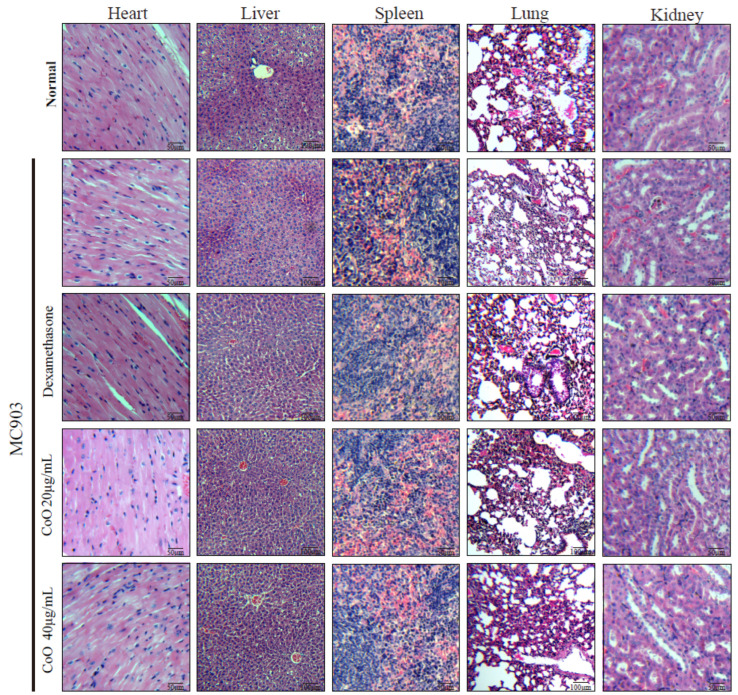
HE-stained image of mouse internal organs.

**Figure 5 nanomaterials-12-00638-f005:**
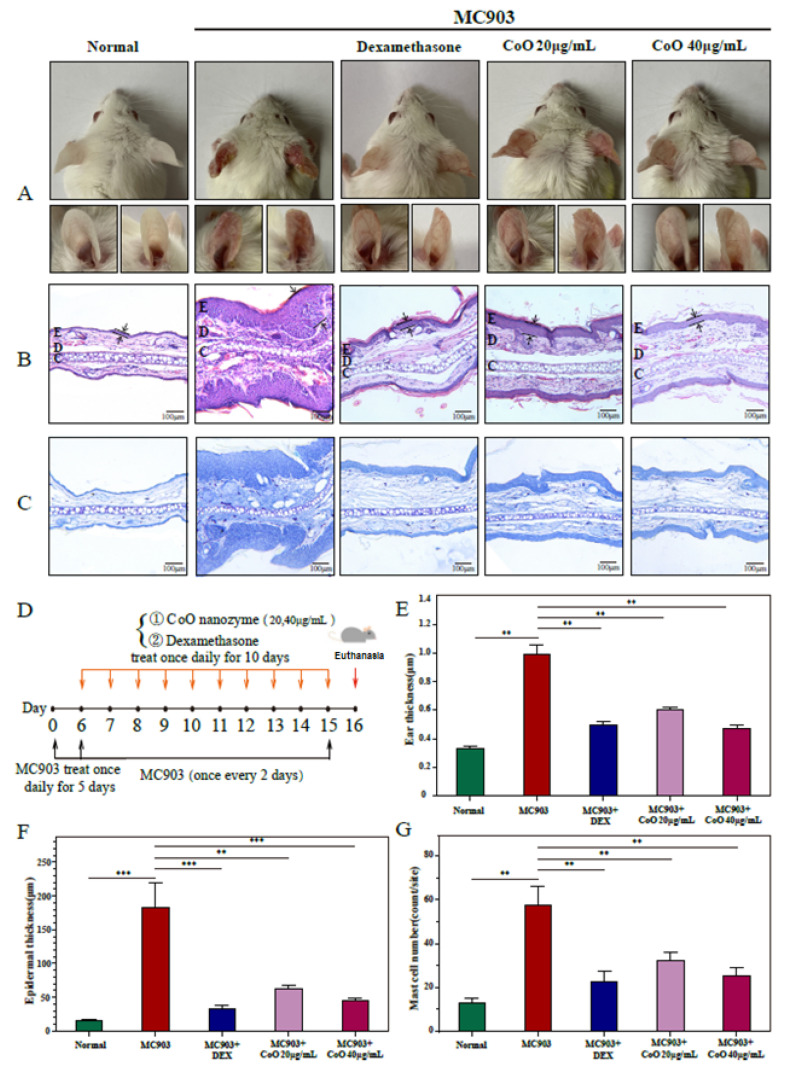
Effects of CoO nanozymes on MC903-induced AD-like skin lesions of mice. (**A**) Representative gross appearance of AD animal models with CoO NPs topical treatment on day 16. (**B**) Ear tissues were stained with HE, ×100. E: epidermal layer; D: dermal layer; C: cartilaginous layer. The epidermal thickness of each group was depicted as intervals between two arrows. (**C**) Ear tissues were stained with toluidine blue, ×100. (**D**) Schematic of the experimental design for the ear skin damage model and subsequent CoO NPs administration. (**E**) Effects of CoO NPs on ear thickness in AD mice. (**F**) Epidermal thickness was analyzed in HE-stained tissue. (**G**) The number of mast cells was analyzed in the toluidine blue-stained sections. Data represented as means ± SDs of three animals (N = 3); ** *p* < 0.01, *** *p* < 0.001.

## Data Availability

Not applicable.

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
