# Peer review of "CoO Nanozymes with Multiple Catalytic Activities Regulate Atopic Dermatitis"

_nanomaterials, 2022, doi:10.3390/nano12040638_

Round 1
Reviewer 1 Report
Please check the attachment. Thank you.

Author Response
Since only one file can be uploaded, we have combined replies, revised manuscripts, and supplementary materials into a single file. Please see the attachment.

Reviewer 2 Report
Please find the attachment.

Author Response

(The authors gave the same response as above.)

Round 2
Reviewer 1 Report
The authors satisfied all my comments.
I suggest to publish it in Nanomaterials.
Author Response
Thanks for your comments, we have partially revised and supplemented the manuscript based on comments from another reviewer, details can be found in the attachment.

Reviewer 2 Report
Please see the attachment.

Author Response
We have revised the manuscript based on your comments, adding the latest methods of nanozyme preparation in Introduction, while adding the content of the comparison of the properties between nanozymes and natural enzymes. Comparisons with other studies are made in the Results and Discussion, highlighting the inadequacies of our study and directions for further research. The detailed information can be seen in the attachment, we hope this revision can meet your requirements. Thank you.
